# Utilization Pattern of Indigenous and Naturalized Plants among Some Selected Rural Households of North West Province, South Africa

**DOI:** 10.3390/plants9080953

**Published:** 2020-07-28

**Authors:** Abiodun Olusola Omotayo, Peter Tshepiso Ndhlovu, Seleke Christopher Tshwene, Adeyemi Oladapo Aremu

**Affiliations:** 1Food Security and Safety Niche Area, Faculty of Natural and Agricultural Sciences, North-West University, Private Bag X2046, Mmabatho 2745, North West Province, South Africa; 25301284@nwu.ac.za; 2Indigenous Knowledge Systems (IKS) Centre, Faculty of Natural and Agricultural Sciences, North-West University, Private Bag X2046, Mmabatho 2745, North West Province, South Africa; Tshepiso.Ndhlovu@nwu.ac.za; 3Department of Agricultural Economics and Extension, Faculty of Natural and Agricultural Sciences, North-West University, Private Bag X2046, Mmabatho 2745, North West Province, South Africa; 16969669@nwu.ac.za

**Keywords:** ethnobotanical survey, food security, fruits, grains, legumes, vegetables

## Abstract

Globally, a substantial proportion of existing indigenous and naturalized foods are often neglected thereby narrowing the food-base available to humans. The current study explored the use and contribution of indigenous and naturalized plants to the households’ food-pool among 12 communities in the North West Province of South Africa. An ethnobotanical survey was conducted among 133 rural households across the 12 selected communities from the four district municipalities in North West Province, South Africa. We analyzed the utilization patterns for 31 selected indigenous and naturalized plants (grains, fruits, and vegetables) among the 133 households using two ethnobotanical indices. In reference to the checklist of 31 plants, the sampled households utilize approximately 94% (29) as staple foods, beverages, fruits and fodder. *Sorghum bicolor, Vigna unguiculata, Amaranthus* sp., *Sclerocarya birrea*, *Persea americana,* and *Mimusops zeyheri* were among the top-six plants based on the Relative Frequency of Citation (RFC, 40–83%). In terms of the different crop-types, *Sorghum bicolor* (grain), *Amaranthus* sp. (vegetable), and *Sclerocarya birrea* (fruit) were the top-ranked plants based on the Species Popularity Index (SPI, 0.53–0.83) among the participants. Overall, there is a need for a renaissance of indigenous and naturalized plants, which has the potential to encourage rural farmers to further embrace the cultivation of these plants on a larger scale so as to enhance food security in the rural communities.

## 1. Introduction

Since time immemorial, indigenous plants have been an essential part of the human diet [1]. Currently, the majority of rural households depend on various natural resources to sustain their livelihoods especially as a way of combating malnutrition and hunger [2,3]. Several studies including ethnobotanical surveys have shown that indigenous plants continue to play an essential role in the livelihoods of rural communities [4,5,6,7,8,9,10,11]. However, the value of these indigenous plants in agriculture as sources of food and medicine for the rural communities is often overlooked. Mabhaudhi et al. (2019) [6] and Ateba et al. (2012) [12], for instance, indicated that the displacement of indigenous plants by a few major ‘exotic’ crops has inevitably contributed, in part, to the limited successes of the global food systems, especially in under-developed regions. Thus, promoting the consumption of the indigenous and naturalized grains, legumes, and other local staple foods is becoming a top priority among policy-makers as a means of alleviating food insecurity and poverty in developing countries [13,14].

South Africa remains one of the major hotspots of biological and ethnic diversity in the Southern African region [15,16]. This rich bio-diversity provides exciting prospects for scientific research and innovations that could be of major cultural and commercial significance nationally and globally [17]. However, indigenous edible plants have been receiving limited attention in terms of research and commercialization [6,18,19,20]. South Africa is among the few countries in Africa that have been pursuing global agro-industrial food systems for decades [6]. Therefore, mainstreaming of indigenous and naturalized plants into the existing food systems in South Africa would help to support the rural inhabitants to diversify their landscape in an economic and all-round sustainable manner. This could translate into the ability of households to feed their immediate family and provide nutritious food at local markets for income generation [21].

As highlighted by Ateba et al. (2012) [12], there has been an increasing interest with regard to Batswana (major ethnic group in North West Province) indigenous knowledge on food and medicinal plants for both sustainable livelihoods and commercial purposes. Hence, the present study explored the utilization pattern of 31 selected South African indigenous and naturalized plants among rural households in the North West Province. The objective was approached by using the existing literature to generate a checklist of 31 indigenous and naturalized plants recognized as common and important in South Africa. The current study addresses the following research questions:Which indigenous and naturalized plants are known by the rural households?How are the selected indigenous and naturalized plants utilized among the rural households?

## 2. Materials and Methods

### 2.1. Study Area

In order to generate baseline information on the use of indigenous and naturalized plants in North West Province, the study was conducted across 12 communities from the four districts (Figure 1 and Table 1). The selection of the 12 communities was based on their rural nature and cultures as well as the low socio-economic status of the residents. The North West province lies between 22° and 28° longitude east of the Greenwich meridian. The province shares boundaries with Northern Cape, Free State, Gauteng and Limpopo as well as an international border with Botswana [22].

The province has a total surface area of 104,882 km^2^. It has a total population of 3,979,000 inhabitants consisting of 89.8% black, 7.3% white, 2% coloured, and 0.6% Indian/Asian [23]. The province is also characterized by great seasonal and daily variations in temperatures ranging from 17 to 31 °C in summer and 3 to 21 °C in winter, with an annual rainfall of 360 mm between October and April [24]. The main language spoken is Setswana (63.4%) and Northern Sotho (2.4%) which is the least spoken language among the 11 South African official languages. Batswana is the main ethnic group in the province [25].

### 2.2. Ethnobotanical Survey

The current survey was guided by previous ethnobotanical studies [5,26,27]. According to Department of Agriculture Forestry and Fisheries [28], “South Africa’s indigenous food crops refer to food crops that have their origin in South Africa. Added to these crops are those that were introduced into the country and are now recognized as naturalized or traditional crops of South Africa.” On this basis, we compiled a list of 31 indigenous and naturalized plants recognized as important and popular based on information from different sources such as the Department of Agriculture Forestry and Fisheries [28], Mashile et al. (2019) [8], Aremu et al. (2019) [29], van Wyk [17], and the government agency website (agribook.co.za/horticulture/indigenous-food-crops). In addition, a photo album of the 31 selected plants was compiled to aid visual recognition during the survey; the pictures were derived from the aforementioned literature and where absent, we consulted reliable websites.

The study was preceded by a pilot test and the botanical names were verified using the Plant List (http://www.theplantlist.org/). Data were collected from September to November 2019 with a semi-structured questionnaire after a successful pilot test. We used face-to-face interviews to engage the 133 households. The questionnaire also included the local names of the selected plants.

For the current study, we targeted participants heading their households from the 12 communities namely: Lomanyaneng, Makhubung, Montshioa, Ventersdorp, Ikageng, Boikhutsong, Hebron, Itireleng, Kgabalatsane, Schwazie-Reneke, Stella, and Vryburg. From each community, 15 questionnaires were randomly administered to participants from different households. However, 11 households from all the communities, with the exception of Montshioa (with 12), fully completed the questionnaires (Table 1). As a result, we had a total of 133 participants with a valid questionnaire which was used in this study. The sample size was robust and quite representative of the selected rural communities. The ages of the participants ranged from 18–80 years. They encompassed different socio-economic strata and had varying degrees of knowledge of indigenous plants.

### 2.3. Data Analysis

The data collected was analyzed using SPSS (version 26) software. The ethnobotanical indices including the Relative Frequency of Citation (RFC, %) and Species Popularity Index (SPI) were determined as described below.

#### 2.3.1. Relative Frequency of Citation (RFC)

Relative Frequency of Citation (RFC) indicates the relative importance of each plant in the study area, without taking into account the use-categories [30]. It was calculated using the formula below:(1)$$\mathrm{RFC}=\frac{\mathrm{FC}\;}{N}\times 100$$
where FC = frequency of quotation/mention (the number of participants indicating the use of the plant), N = total number of participants.

#### 2.3.2. Species Popularity Index (SPI)

The SPI was analyzed using the matrix method derived by De Beer and van Wyk [31]. This approach offers a quantified measure of information on the ranking. The matrix method is based on three questions, which rate the species popularity. The three questions included: (1) Do you know the plant species? Score 1 (yes) or 0 (no). (2) Do you have any name for that plant species? Score 2 (yes) or 0 (no). (3) What is its use? Score 3 (yes) or 0 (no). Based on the above information, a matrix was generated and we calculated the SPI by ratio of total species score divided by maximum possible score [31]. The SPI was calculated separately for the three (grains, vegetables and fruits) categories of the indigenous and naturalized plants.
(2)$$\mathrm{Species}\;\mathrm{Popularity}\;\mathrm{Index}\;(\mathrm{SPI})=\frac{\mathrm{Total}\;\mathrm{score}\;\mathrm{for}\;a\;\mathrm{species}\;}{\mathrm{Maximum}\;\mathrm{possible}\;\mathrm{score}\;(798)}$$

### 2.4. Ethical Approval

The ethical clearance (certificate no: NWU-01243–19-S9) for the research was approved as a low risk by the ethics committee of the Faculty of Natural and Agricultural Sciences, North West University, South Africa. The permit to access the study area was granted by the North West Provincial Department of Rural, Environment and Agricultural Development (READ), South Africa, before the administration of questionnaires in the study area.

The ethnobotanical survey was conducted with the full consent of the participants. We provided details of the research and expectation from the researchers and participants. These included the rule of voluntary participation and withdrawal of the participant at any given time. During the course of this study, the principle of privacy, autonomy, dignity, and respect (Ubuntu) was handled with diligence between the researchers and participants.

## 3. Results and Discussion

### 3.1. Demographic Overview of the Participants

The age distribution shows that the majority (37%) of the participants were 51–60 years old followed by 41–50 year-old (27%) and 31–40 year-old (19%) individuals. On the other hand, the lowest categories were individuals aged above 71 years (9%) and 20–30 years (8%). In addition, the study indicates that most of the participants acquired formal education at primary (33%), secondary (32%), and tertiary level (30%), while 5% of the participants had no formal education. Furthermore, the gender distribution of the participants indicates that 53% of the households were headed by females compared to 47% for male-headed households in the study. The dominance of female-headed households is becoming a common trend in South Africa, and has been corroborated by existing research [32,33].

In terms of household size, the majority (45%) of households had 6–7 individuals. Similar results of a relatively high number of individuals have been observed in many rural areas in South Africa [32,34,35]. In the current study, 89% of the participants had detailed knowledge of indigenous and naturalized plants while 11% had limited knowledge. Parents (38%) and community members (28%) were the major sources of the indigenous knowledge on the indigenous and naturalized plants.

### 3.2. Ethnobotanical Indices and Use-Categories for the 31 Selected Indigenous and Naturalised Plants

In the current study, the analysis shows that the participants identified 31 indigenous plants that are commonly utilized in the study area (Table 2). Higher RFCs were reported for *Sorghum bicolor* (83%), *Vigna unguiculata* (53%), *Amaranthus* sp. (53%)*, Sclerocarya birrea* (53%), *Persea americana* (53%), and *Mimusops zeyheri* (40%), while others generally had low RFCs (<20%).

Furthermore, we categorized the selected indigenous and naturalized plants into three groups namely: grains, vegetables, and fruits in order to establish their SPI within each group (Table 3). On this basis, *Sorghum bicolor* had the highest SPI (0.834) whereas *Tylosema esculentum* and *Vigna radiata* (SPI = 0.007) were the least ranked among the indigenous and naturalized grains. *Amaranthus* spp. and *Sclerocarya birrea* (SPI = 0.533) were the top-ranked indigenous and naturalized vegetables and fruits, respectively. On the other hand, *Dovyalis caffra* was the least (SPI = 0) ranked indigenous and naturalized fruit among the participants. In the present study, 56% of plants were domesticated while 46% were collected from the wild. However, one of the plants was currently sourced from the wild and also domesticated.

As indicated by Ateba et al. (2012) [12], the North West Province has an enormous richness and diversity of indigenous and naturalized plants. We generated five use-categories for the selected 31 plants (Figure 2). In some cases, different parts of the same plant had different uses. In the current study, the staple indigenous and naturalized food category recorded the highest use (50%), while snacks and fodder had the lowest (3%) uses among the rural households. According to Willett et al. (2019) [36], the targets for healthy human diets are composed largely of vegetables and fruits, whole grains, legumes, nuts, and unsaturated oils, together with seafood and poultry. Thus, the dietary needs of humans can be met through the consumption of diverse indigenous and naturalized plants known to the participants in the study area. Moreover, Bvenura and Sivakumar [21] indicated that the nutritional composition of indigenous and naturalized fruits and vegetables contribute positively to the human diet. The daily consumption of indigenous and naturalized plants has potential to enhance proper growth, good health and well-being [6,37]. However, studies have confirmed that people with high income and formal education attainment mostly show negative attitudes towards indigenous and naturalized plants [38]. Therefore, indigenous and naturalized plants have been gradually losing their importance when compared to exotic food crop varieties over the last few decades [39,40].

Mabhaudhi et al. (2019) [6] indicated that many under-utilized indigenous plants contain high nutritional value and could improve the nutritional status of many impoverished individuals. In the current study, *Pennisetum glaucum* was one of the plants that had three use-categories, namely beverages, staple, and livestock feed. It is evident that *Pennisetum glaucum* is largely cultivated in arid and semi-arid regions of Africa and India as grain for human consumption and forage for livestock [41]. Furthermore, in Sub-Saharan Africa, *Pennisetum glaucum* constitutes both a unique ecological heritage and a critical food security component among millions of small-scale farmers. Kucich and Wicht [42] indicated that the nutritionally superior indigenous crops have gradually been displaced by cash crops that do not properly serve poor rural communities, thereby placing rural populations especially children at a higher risk of starvation and poverty. In South Africa, indigenous plants especially leafy vegetables have the potential to mitigate malnutrition due to their high nutritional content [43]. Several studies have established the important role and contribution of indigenous and naturalized plants to the economy and to livelihoods of households in rural areas [5,44,45,46,47].

### 3.3. Uses and Benefits of Indigenous and Naturalised Grains

*Sorghum bicolor* was the most widely used indigenous and naturalized grain among the households (Table 3). Globally, *Sorghum bicolor* is the dietary staple food of more than 500 million people in more than 30 countries [36,48]. In sub-Saharan Africa and Asia, small grain cereals such as sorghum and millets contribute significantly to food security, nutrition and health [49,50]. In rural areas, the traditional cereals such as sorghum and millets are normally boiled into porridges (thick) or gruels (thin) for consumption. Furthermore, small grains have the greatest amount of untapped potential and are predominant in South Africa [28]. *Sorghum bicolor*, for instance, is the most important naturalized grain for household food security in South Africa [50].

In the current study, *Vigna unguiculata* ranked second highest (SPI = 0.532) amongst the indigenous and naturalized grains. This result is not surprising as *Vigna unguiculata* is rated among the indigenous foods with high nutritional value (protein, energy, fiber, vitamins, and minerals), broad social acceptance, and is used as a relish accompanying staples such as ‘pap’ (maize meal) and rice [51]. *Vigna unguiculata* is a staple legume in sub-Saharan Africa and other semi-arid warm tropics and sub-tropics [52]. The participants further mentioned that the availability of the *Vigna unguiculata* provides sufficient resources in the form of fresh as well as dried seeds. According to Ehlers and Hall [53], *Vigna unguiculata* is a widely-adapted, stress-tolerant grain legume, vegetable, and fodder crop grown in an estimated 7 million hectares in warm to hot regions globally.

### 3.4. Uses and Benefits of Indigenous and Naturalised Vegetables

Many *Amaranthus* species are well-known for their medicinal value and as a rich source of vitamins [54,55]. They are easy-to-grow herb that are widely distributed globally. The study revealed that amongst the 11 indigenous and naturalized leafy vegetables, *Amaranthus* sp. was the most popular species (Table 3). Mnkeni et al. (2007) [54] indicated that many cultivars of *Amaranthus* are highly nutritious but are hardly consumed as food in many parts of South Africa. Vegetables are a critical source of many micronutrients, including pro-vitamin A for the prevention of night blindness. High intake of vegetables reduces blood pressure and is associated with lower risk of type 2 diabetes [36,56,57].

Despite the abundance of traditional leafy vegetables, they remain under-exploited and under-utilized due to various limitations [58]. In South Africa, there is a stigma towards consuming indigenous leafy vegetables, especially the traditional leafy vegetables (morogo) [59]. As observed by Vorster et al. (2008) [60], the use of indigenous leafy vegetables in rural communities has reached a low point, as many have labeled such dishes as poverty food. Hence, there is a general decline in the consumption of indigenous leafy vegetables mainly due to inadequate knowledge on their benefits. Furthermore, Smith and Eyzaguirre [58] reported that exotic vegetables such as cabbage and spinach have almost been completely replaced African leafy green vegetables in local areas. van Rensburg et al. (2014) [59] articulated that indigenous leafy vegetables are considered to be a rich source of food and medicine for humans and animals. Likewise, Welcome and van Wyk [61] reported that indigenous leafy vegetables still contribute greatly to food security in households.

The consumption of leafy vegetables is also linked to the traditions and dietary patterns of each ethnic and socio-economic group [21,62,63]. Additionally, ethnicity and geographical location strongly influence the choice and consumption of indigenous and naturalized leafy vegetables [64]. Since some parts of the North West Province experience low rainfall annually, the occurrence of indigenous leafy vegetables is rare and results in low consumption thereof [24].

### 3.5. Uses and Benefits of Indigenous and Naturalised Fruits

As in other provinces in South Africa, several local communities still consume a wide range of wild and semi-wild fruit species in the North West Province [12]. From the 13 listed indigenous and naturalized fruits, *Sclerocarya birrea* was the most popular among the selected households (Table 3). The consumption of indigenous fruits among the households may be attributed to the presence of a high number of fruit trees in the study area. Fruits are an important source of nutrition with diverse physiological functions and health benefits [36]. Furthermore, Schreckenberg et al. (2006) [65], Aremu et al. (2019) [29], and Omotayo and Aremu [66] indicated that the potential of indigenous fruits is often not recognized either at national or international level as an efficient and affordable nutrient deficiency reduction strategy. In addition, indigenous and naturalized fruits usually contribute to food and nutrition security, health, and income generation of rural communities [67]. They are well-adapted to their local environments and often survive under drought conditions even when staple crops fail, and as a result serve as an emergency food supply during times of food shortage [68]. Likewise, Mashile et al. (2019) [8] indicated that food shortages make indigenous and naturalized fruits important food supplements in rural communities.

According to Mengistu and Hager [69], different plants are used by different communities and the importance of species depends on local practices. In addition, the wide range of indigenous fruit trees available in many areas can enable local farmers to meet their household needs such as food, nutrition, and medicine [19]. Particularly, the daily consumption of indigenous and naturalized fruits can substantially lower the risk of mortality especially from several non-communicable diseases [21,42,67].

## 4. Conclusions

It is evident from the current findings that rural households in the selected areas of the North West Province still consume indigenous and naturalized plants. The sampled households still consume 94% of the 31 indigenous and naturalized plants, especially *Sorghum bicolor, Vigna unguiculata, Amaranthus* sp., *Sclerocarya birrea*, *Persea americana* and *Mimusops zeyheri*. In addition, *Sorghum bicolor, Amaranthus* sp., and *Sclerocarya birrea* were the top-ranked indigenous and naturalized grain, vegetable, and fruit, respectively. Based on the low RFC and SPI value for most of the plants, there is a need for greater awareness of the numerous benefits associated with the consumption of indigenous plants for the food security and sustainability of rural households. Wider acceptance of indigenous plants will help reduce over-reliance of rural households on the few exotic food varieties, while supporting and enhancing national food self-sufficiency as well as economic sustainability in South Africa. However, the fragile nature of oral-based knowledge and loss of information highlights the importance of more conscious effort aimed at documenting and creating reliable inventory for these indigenous and naturalized plants. Particularly, it will be essential to document important knowledge (e.g., preparation methods especially for beverages and health drinks) associated with these indigenous and naturalized plants. Furthermore, there is a need for a renaissance in the use of indigenous and naturalized plants, which will focus on disadvantaged communities and potentially empower rural farmers to produce more indigenous plants leading to better livelihoods of South African rural communities.

## Figures and Tables

**Figure 1 plants-09-00953-f001:**
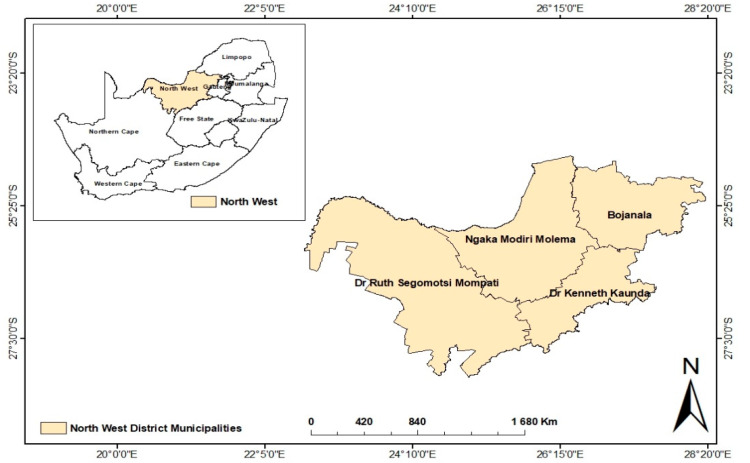
Study sites were located across the four district municipalities of the North West Province, South Africa.

**Figure 2 plants-09-00953-f002:**
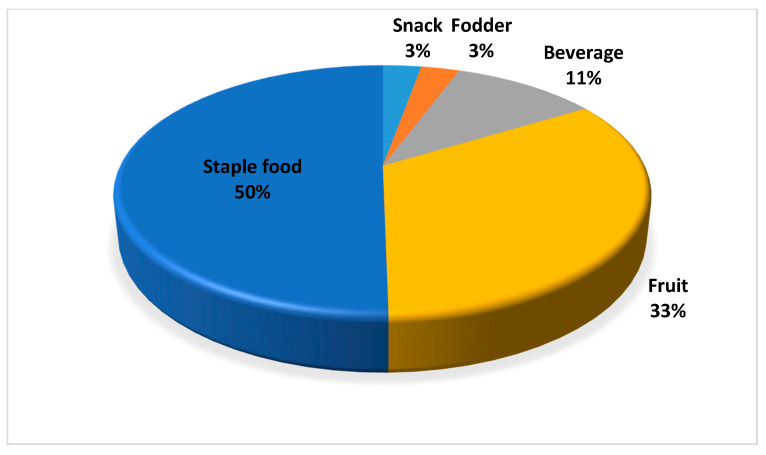
Use-categories for indigenous and naturalized plants among the 133 rural households in North West Province, South Africa (n = 36).

**Table 1 plants-09-00953-t001:** Study sites (villages and townships) in the four districts in North West Province, South Africa.

	District Municipality	Community	No of Administered Questionnaires	No of Properly Filled Questionnaires
1.	Ngaka Modiri Molema	Lomanyaneng	15	11
		Makhubeng	15	11
		Montshioa	15	12
2.	Dr Kenneth Kaunda	Venterdorp	15	11
		Ikgageng	15	11
		Boikhutsong	15	11
3.	Bojanala Platinum	Hebron	15	11
		Itereleng	15	11
		Kgabalatsane	15	11
4.	Dr Ruth Segomotsi Mopati	Schwazie-Reneke	15	11
		Stella	15	11
		Vryburg	15	11
		Total	180	133

**Table 2 plants-09-00953-t002:** Ethnobotanical information on indigenous and naturalized plants consumed by rural households in the North West province, South Africa. ^^ = naturalized plants; ^$^ Common/vernacular name (A = Afrikaans, E = English, Z = Zulu, Ns = Northern Sotho, Ss = Southern sotho, Ts = Setswana, Tso = Xitsonga, V = VhaVenda, X = Xhosa); * O = Occurrence (W = Wild and D = Domesticated); Ethnobotanical Index used, FC = Frequency of Citation and RFC = Relative Frequency of Citation).

Scientific Name &	^$^ Common/Vernacular Name	Plant Part Used	Usage			
Family Name				* O FC		RFC (%)
*Amaranthus* sp. **Amaranthaceae**	Thepe (Ts); Amaranthus (E); Infino (Z)	Leaves	Staple food	W	71	53
*Annona senegalensis* Pers. **Annonaceae**	African Custard-apple (E); Custard Apple (E), Isiphofu (Z); Mokamanawe (Ts); Motlepe (Ns)	Fruit	Fruit	W	18	13.5
*^^ Bidens pilosa* L. **Asteraceae**	Mokolonyane (Ts); Blackjack (E); gewone knapsekêrel (A); Umhlabangubo, Uqadolo(Z)	Leaves	Staple food	W	15	11.2
*^^ Brassica juncea* (L.) Czern. **Brassicaceae**	Ethiopian Mustard (E)	Leaves	Staple food	D	0	0
*Cajanus cajan* (L.) Millsp. **Leguminosae/Fabaceae**	Pigeon bean (E); Dinawa (Ts)	Seeds & leaves	Staple food	D	26	19.5
*Carissa macrocarpa* (Eckl.) A.DC. **Apocynaceae**	Natal plum, big num-num (E); grootnoem-noem (A); Amatungulu (Z)	Fruit	Fruit	D	1	0.7
*^^ Chenopodium album* L **Amaranthaceae**	Fat hen (E)	Leaves	Staple food	W	8	6
*^^ Citrus* sp. **Rutaceae**	Lemon (E)	Fruit	Fruit	D	59	44
*Cleome gynandra* L **Cleomaceae**	Lerotho (Ts); Cat’s whiskers, Cleome, African cabbage (E); Snotterbelletjie (A)	Leaves	Staple food	D	4	3
*^^ Colocasia esculenta* (L.) Schott **Araceae**	Amadumbe, Amadombie, Amadombi, Mufhongwe (Z)	Rhizome	Staple food	D	3	2.2
*^^ Corchorus olitorius* L. **Malvaceae**	Jew’s mallow, wild jute (E); Wildejute (A); Thelele,(Ns); Delele, Gushe (Ts)	Leaves	Staple food	W	14	10.5
*^^ Cucurbita pepo* L. **Cucurbitaceae**	Pumpkin (E); Lephutsi (Ts)	Whole plant	Staple food	D	20	15
*Diospyros lycioides* Desf. **Ebenaceae**	Monkey plum (E); bloubos (A); Lethanyu (Ts); Muthala (V); Monkga-nku (Ss); Mtloumana (Ns); Umbhongisa (X); Umbulwa (Z)	Fruit	Fruit	W	1	0.7
*Diospyros simii* (Kuntze) De Winter **Ebenaceae**	Climbing Star-apple (E); Kraaibessie (A)	Fruit	Fruit	D	2	1.5
*Dovyalis caffra* (Hook.f. & Harv.) Sim **Salicaceae**	Kei-apple (E); Kei-appel (A); Motlhono (Ns); Umqokolo (Z)	Fruit	Fruit	D &W	0	0
*Dovyalis zeyheri* (Sond.) Warb. **Salicaceae**	Wild apricot (E); Wilde-appelkoos (A); umNyazuma (Z); umQokokolo (X)	Fruit	Fruit	W	1	0.7
*Lagenaria siceraria* (Mol.) Standl. **Cucurbitaceae**	Bottle gourd, Calabash (E); Kalbas (A); Moraka (Ns); Segwana (Ts); Iselwa (X, Z)	Whole plant	Staple food	D	9	6.7
*^^ Manihot esculenta* Crantz **Euphorbiaceae**	Muthupula (Ts); Umdumbula Othobola (Z)	Rhizome	Staple food	D	8	6
*Mimusops zeyheri* Sond **Sapotaceae**	Transvaal red milkwood (E); Moepel (A); Mmupudu (Ns); umpushane (Z); Mubululu (V)	Fruit	Beverage	W	53	39.8
*Parinari curatellifolia* Planch. ex Benth. **Chrysobalanaceae**	Bosappel (A); Mmola (Ns); Mbulwa (Tso); Mobola (Ts); Muvhula (V)	Fruit	Fruit	W	4	3
*Pennisetum glaucum* (L.) R.Br. **Poaceae**	Pearl millet (E); Nyalothi, Ntweka, Amabele (Z); Inyawuthi, Muvhoho (V); Babala, Manna (Ts)	Grains	Beverage, staple food and fodder	D	13	9.7
*^^ Persea americana* Mill. **Lauraceae**	Avocado (E)	Fruit	Fruit	D	70	52.6
*Sclerocarya birrea* (A.Rich.) Hochst. **Anacardiaceae**	Marula (E); Morula (Ns); Mufula (V); ukanyi (Ts)	Fruit	Fruit and beverage	W	71	53.3
*Sorghum bicolor* (L.) Moench **Poaceae**	Sorghum (E); Graansorghum (A); Mabele (Ts); Amabele (Z); Amazimba (X)	Grains	Beverage and staple food	D	111	83.4
*Strychnos spinosa* Lam **Loganiaceae**	Spiny Monkey-orange (E); doringklapper (A); Morapa, Nsala (Ts)	Fruit	Fruit	W	8	6
*Tetragonia decumbens* Mill. **Aizoaceae**	Dune spinach (E); Duinespinasie (A)	Leaves	Staple food	D	32	24
*Tylosema esculentum* (Burch.) A.Schreib. **Leguminosae/Fabaceae**	Marama bean (E)	Seeds	Staple food	W	1	0.7
*Vangueria infausta* Burch **Rubiaceae**	Wild-medlar, (E); Wilde mispel (A); Mothwany, Mmilo (Ts); mmilo (Ns); muzwilu, mavelo (V);umvilo (X); umviyo, umtulwa (Z);	Fruit	Fruit	W	15	11.2
*^^ Vigna radiata* (L.) R.Wilczek **Leguminosae/Fabaceae**	Mung bean (E)	Seeds & leaves	Staple food	D	1	0.7
*^^ Vigna subterranean* (L.) Verdc. **Leguminosae/Fabaceae**	Bambara groundnut (E)	Seeds	Staple food and Snacks	D	27	20.3
*^^ Vigna unguiculata* (L.) Walp. **Leguminosae/Fabaceae**	Cowpea (E); dinawa (Ts); imbumba, indumba (Z)	Seeds & leaves	Staple food	D	71	53.3

**Table 3 plants-09-00953-t003:** Ranking of indigenous and naturalized plants utilized among 133 rural households in North West Province, South Africa.

Species by Use	Rank	Species Popularity Index (SPI)
**Indigenous and naturalized grains**		
*Sorghum bicolor*	1	0.834
*Vigna unguiculata*	2	0.533
*Vigna subterranean*	3	0.203
*Cajanus cajan*	4	0.195
*Pennisetum glaucum*	5	0.097
*Tylosema esculentum*	6	0.007
*Vigna radiata*	7	0.007
**Indigenous and naturalized vegetables**		
*Amaranthus* sp.	1	0.533
*Tetragonia decumbens*	2	0.240
*Cucurbita pepo*	3	0.150
*Bidens pilosa*	4	0.112
*Corchorus olitorius*	5	0.105
*Lagenaria siceraria*	6	0.067
*Chenopodium album*	7	0.060
*Manihot esculenta*	7	0.060
*Cleome gynandra*	9	0.030
*Colocasia esculenta*	10	0.022
*Brassica juncea*	11	0.000
**Indigenous and naturalized fruits**		
*Sclerocarya birrea*	1	0.533
*Persea Americana*	2	0.526
*Citrus* sp.	3	0.443
*Annona senegalensis*	4	0.135
*Mimusops zeyheri*	5	0.398
*Parinari curatellifolia*	6	0.030
*Vangueria infausta*	7	0.112
*Strychnos spinosa*	8	0.060
*Diospyros simii*	9	0.015
*Dovyalis zeyheri*	10	0.007
*Diospyros lycioides*	10	0.007
*Carissa macrocarpa*	10	0.007
*Dovyalis caffra*	13	0.000

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
