# Peer review of "Utilization Pattern of Indigenous and Naturalized Plants among Some Selected Rural Households of North West Province, South Africa"

_plants, 2020, doi:10.3390/plants9080953_

Round 1

Reviewer 1 Report

This paper has a valid scientific objective and has the potential to make a useful contribution to the scientific literature on the use of neglected and underutilized food plants by African communities. Moreover, the overall presentation is of a quality consistent with publication.  However, the paper has fundamental and serious deficiencies that undermine the credibility of the methods used and the validity of the results presented.  It is not acceptable for publication without major revision.

The paper has four inter-connected deficiencies:

  1. It is stated (lines 83-85 and elsewhere) that interviews were based on a list of 33 plants compiled from two sources (references 10 and 25) and proceeded using photographs from these sources. Several of the species recorded in Table 2 do not appear in either of these sources. Moreover, reference 10 contains no photographs and reference 25 contains only 18 photographs.
  2. At least 3 of the species listed in Table 2 do not credibly appear in South Africa: Amaranthus acanthobracteatus, Citrus assamensis, Strychnos aculeata. It is essential to better indicate how taxonomic identities were made and to include herbarium specimens (ideally for all species, but certainly for those where identify is questionable).
  3. The manner in which the authors use the word “indigenous” is problematic. This term needs to be defined in the paper. Even if the term is used in a particular manner in South Africa, for publication in a scientific journal it needs to be used in a manner that is consistent with global norms. Consider this statement from the South Africa Department of Agriculture Forestry and Fisheries (reference 25 that the authors cite): “Indigenous food crops refer to food crops that have their origin in South Africa. Added to these crops are those that were introduced into the country and are now recognised as naturalised or traditional crops.” Reference 10 also distinguishes between native and naturalized plants. The authors seem to use “indigenous” to refer to any plant grown by people who are indigenous to South Africa. It is fine for this to be the scope of the paper, but only if the terms are defined clearly and used properly and consistently. If one assumes that all crop plants native to Sub-Saharan Africa are native to South Africa (which they are not), food plants (in addition to the three above) that the authors have misidentified as “indigenous” include: Colocasia esculenta, Cucurbita pepo, Glycine max, Manihot esculenta, Persea americana, Triticum aestivum.
  4.  Triticum aestivum and Glycine max are major agricultural commodites that rank as numbers two and three in terms of global crop value. By any definition these are hard to see as “indigenous plants”.

Crops and wild-harvested plants should be distinguised in Table 2 or elsewhere in the paper.

While the overall quality of English of the paper is high, there are a few deficiencies in grammar and style. In particular some components, most noticably in the Abstract and introductory components of the paper, are excessively and inefficiently wordy with considerable redundancy and repetition of words that makes it difficult to read.

Reviewer 2 Report

This important study encompasses an ethnobotanical survey among 133 rural households across 12 randomly 20 selected communities from the four district municipalities of North West province, South Africa. The authors seek to document the household food pool by means of RFCs and SPIs. However, the study is limited to 33 plants previously selected based on two other studies. The authors correctly argue that a huge amount of diverse indigenous food plants were replaced by only a few “exotic” crops, leaving only small potential for a balanced diet among rural people. The results and discussion are nicely prepared, but lack some important aspects of review of existing literature regarding the use of the selected plants as food sources. However, more lack of information is present in the methods section as explained below in the comments.

  1. Line 20: An ethnobotanical survey (missing word)
  2. Line 39: displacement (instead of displayment)
  3. Line 43: some of the sentences in the introduction lack prepositions. This sentence starts with “Although, …” but the second part of the sentence is missing (“Although, … ,…”), therefore making no sense.
  4. Line 47: “South Africa remains one of the major hotspots of biological and ethnic diversity in the Southern African region [15].” This paper/reference by Ben-Erik van Wyk is from 1997. Please, update if you say “remains”. Or say: …was previously identified as one of the major hotspots…
  5. Line 50: “However, indigenous edible plants have been receiving limited attention in terms of research and commercialization [17, 18].” Update with newer references to show that they still have been receiving limited attention in research and commercialization.
  6. Introduction/Methods:
    To me, it is not clear why you have chosen these 33 indigenous plants, instead of surveying for the totality of plants locally consumed for food. Any information on how this predefined list of 33 indigenous plants was generated is missing, just the sources are given in the methods section (based on previous studies). If you use this strategy, it is important to justify scientifically and reasonably why these plants were selected and others were excluded, e.g. because of results of a preliminary survey/study that was conducted in the region. Are these all of plants mentioned in the other studies or why did you only pick 33 from these publications? What is the research rationale here? How did you secure that all of these plants are really endemic and not originally imported to the region, the same way as those plants you referred to as “exotic” crops?
  7. Introduction:

It would be important to define “indigenous crops” and “exotic crops”, because some of the plants on the list are also cash crops locally and globally. Mention the most important exotic crops, because for readers from other parts of the world, it is not automatically clear what crops are considered “exotic” in South Africa.

  1. Figure 2 shows use categories (fodder, snacks, beverages, staple foods, fruits) that differ from the SPI categories (grains/vegetables/fruits). Information on sorting of uses into use categories needs to be given in the methods section as standalone data analysis subsection.
  2. Discuss why not all questionnaires were properly filled? Did you conduct the interviews yourselves or did you simply hand them out and collected them again later? The procedure should be explained more detailed.
  3. Please, provide a copy of the questionnaire in the Annex/Supplementary material (in English language and if applicable, in local language/Batswana).
  4. If available, it would be meaningful to state the number of family members per household (percentage ranges in the results section). If you don’t have this information, it is also fine, but I think it would contribute to the quality of the paper.
  5. Line 107: Please, provide a formula for the SPI.
  6. Results section and methods formula:
    It is more conclusive to understand and read if the RFCs are given as percentage values throughout the paper.
  7. Table 2, Heading: “The botanical names were verified using the Plant List (http://www.theplantlist.org/).” Move this sentence to the methods section.
  8. Line 153: This sentence does not make sense (“…, low to moderate consumption of seafood and poultry)
  9. It would have been nice to know more about the ways of preparation, especially for the beverages. This information is missing in table 2. If available, please add or incorporate into your questionnaire design for future studies.
  10. Table 3: Table 3 is now called Table 2. Please, correct the Table #.
  11. Line 150: were identified (instead of well identified)
  12. Line 163: “However, studies have confirmed that people with high income and formal education attainment mostly show negative attitude towards indigenous plants utilization. Therefore, indigenous plants are gradually losing their importance when compared to exotic food crop varieties over the last few decades [33, 34]. ” For the questionnaire setup, it would have been nice to know what the perception of the participants regarding the indigenous plants is. Do they value them more or do they value them less than the exotic food crops? What is the ratio/percentage of them using exotic crops compared to indigenous crops for food purposes?
  13. Discuss how plants most often cited compare to other regions and countries/studies published in terms of use and use categories. For examples, Pennisetum glaucum seems to be used in many ways (3 use categories). Is this unique to the region or has it been described somewhere else before? (literature review)

Reviewer 3 Report

Please check word use, and revise some of the sentence structure. Not sure why the survey was administered to a small group in so many communities (n=ca.10 per community) but perhaps this is a pilot project. Perhaps  you can discuss this. One final question is why 94% of the plants are used if the plants are all recognized grains, fruits and vegetables. My understanding is that this would be 100%. Please make this clear.  Nevertheless, it is very interesting. 

Round 2

Reviewer 1 Report

Although the revised manuscript addresses some of the deficiencies of the previous versions it is not complete or sufficient in this regard. The paper is still not acceptable for publication.

  1. Identification of two of the species remains problematic:                                                                                  Citrus assamensis is not credibly present in South Africa. CAB Abstracts does contain 34 publications relating to this species, all of them from Northern India. The English name provided by the authors is “lemon”, which might refer to Citrus X limon. Then again it might refer to other Citrus hybrids with English names such as: imperial lemon, ponderosa lemon, Meyer lemon, hirami lemon, volamer lemon or ichang lemon (cf. Citrus in Wikipedia). Based on www.fondazioneslowfood.com and similar links in Google this could be the Cape Rough Skinned Lemon (Citrus jambhiri). Unfortunately, The Plant List.org and Worldfloraonline.org recognizes the name C. jambhiri as Ambiguous. This is not as problematic as it sounds. The identification of Citrus assamensis can stand only if the authors provide a herbarium specimen and the name of a recognized authority who confirms the identification of the specimen. However, in reality cultivated hybrids (particularly Citrus) are notoriously problematic and without genome data a lot of identities are just approximate guesses.  The simplest solution is to call the plant Citrus sp. or Citrus cf. x limon.                                                                                                                                                              Strychnos spinosa. This is a more believable name than what appeared in the earlier draft. This is a widespread South African species. However, the vernacular names provided by the authors do not correspond to what is in published sources (see PlantZAfrica.com (pza.sanbi.org)) for S. spinosa. Rather the vernacular names the authors provide correspond to Strychnos cocculoides (see also PlantZAfrica.com). Continued identification of this material as S. spinosa would need to be supported with a herbarium specimen and the name of a recognized authority who confirms the identification of the specimen and/or appropriate references.
  2. The authors have not addressed the problem with how they use the word “indigenous”. This term is still not properly defined in the paper. Inclusion of the statement from the SA Department of Agriculture Forestry and Fisheries only further shines light on the authors’ lack of understanding of this issue. A simple solution would be to change any use of the word “indigenous” to “indigenous and naturalized” [including in the title of the paper] and to include a column (or some other notation) in the table that distinguishes which species are naturalized
  3. Continued inclusion of Triticum aestivum and Glycine max in this paper without providing any justification is unacceptable. Inclusion of these two major crop commodities contributes to an overall incoherence and intellectual inconsistency of this paper. Consider inclusion of these two crops in relation to this statement on lines 41-44 of the paper:                                                                                                                                                   “Mabhaudhi et al. [8] and Ateba et al. [12] for instance, indicated that the displacement of indigenous crops by a few major ‘exotic’ crops has inevitably contributed, in part, to the limited successes of the global food systems, especially in the under-developed regions.”                                                                                                                    Wheat and soybeans are the ultimate in major ‘exotic’ crops. How can they be “indigenous” and “major ‘exotic’ crops” at the same time?   This inconsistency in logic must be corrected!

Reviewer 2 Report

Thanks for the good revision which significantly improved the manuscript. Please, have a look at my comments below:

  1. Line 100:
    Face-to-face interviews were conducted in 133 selected household.
  2. Thanks for defining “indigenous plants”. Still, to me, the definition is still a bit contradictory. “South Africa’s indigenous food crops refer to food crops that have their origin in South Africa. Added to these crops are those that were introduced into the country and are now recognised as naturalised or traditional crops of South ” It says that those exotic crops introduced are added to the indigenous food crops, which means that all exotic plants are indigenous, traditional plants if there is no condition in the definition. Or does this definition mean that there are those crops that are indigenous and then there is a second group which is different (not added) which are crops there were introduced to South Africa? For example, soy bean (Glycine max) was introduced to South Africa in 1903. To me, this is clearly an exotic plant. But still, you refer to it as indigenous food crop. When does a plant become indigenous? After a certain amount of years when it is grown? To me, this is still not clear.
  3. Fidelity level:
    The formula is not correct. The fidelity level is used for identifying the most preferred species for certain treatments (in medicine) or uses (in general). It is displaying the importance of a species for a certain purpose in a community/region. If done correctly, the formula would be the same, but Iu does not represent the total number of participants in the survey, but the total number of respondents who mentioned the same plant for any ailment/use. lp on the other hand is not the number of participants using a particular plant, but the number of respondents who reported the utilization of medicinal plants for a specific main ailment/use (e.g. see here: Friedman, Z. Yaniv, A. Dafni, and D. Palewitch, “A preliminary classification of the healing potential of medicinal plants, based on a rational analysis of an ethnopharmacological field survey among Bedouins in the Negev Desert, Israel,” Journal of Ethnopharmacology, vol. 16, no. 2-3, pp. 275–287, 1986., but it is also in so many publications and books). If you don’t put this value into relation with the use categories you allocated, then you cannot use the fidelity level. What I was requesting was the RFC = FC/N*100, which is basically what you have defined as Fidelity Level. I would suggest deleting the FI from your manuscript and using the percentage values for the RFC (you already calculated these as your “FI”).
  4. After taking a look at your nicely prepared questionnaire, I really wonder, why did you not include all of the data gathered in this study as it belongs to this study (same questionnaires and participants)? Why did you leave out publishing information on the socio-economic background of the participants (education, religion, primary occupation, source of knowledge)? Same goes for the personal opinion on plants, such as advantages, adverse effects, use as medicine, financial return of indigenous plants, perception of indigenous plants, methods of preparation. This information would significantly improve the manuscript, especially if the data is already collected.

Round 3

Reviewer 1 Report

The authors have addressed the major problems in previous drafts of the paper and it is publishable after minor revision.

The inclusion of a notation to identify naturalized species is a  helpful step forward.  However, few of the naturalized species are identified correctly.  I have quickly gone through a few plants with which I have some familiarity as to origin.  The authors need to check every plant on the list against authoritative sources to verify its origins.   In addition Vigna radiata is incorrectly identified as cowpea.  This is most known as mung bean (or green gram) in English.  I do not know if the vernacular names included in the paper are also erroneous.

Here a correct origins of some of the species listed:

Bidens pilosa  - Americas

Chenopodium album - Eurasia

Citrus - Asia

Colocasia esculenta - Asia

Cucurbita - Americas

Manihot esculenta - South America

Vigna radiata - Asia

Reviewer 2 Report

It is OK now, good work!

Author Response

Thank you helping to improve our manuscript. We sincerely appreciate your input.